# Diagnosis of Pancreatic Ductal Adenocarcinoma by Immuno-Positron Emission Tomography

**DOI:** 10.3390/jcm10061151

**Published:** 2021-03-10

**Authors:** Ruth González-Gómez, Roberto A. Pazo-Cid, Luis Sarría, Miguel Ángel Morcillo, Alberto J. Schuhmacher

**Affiliations:** 1Molecular Oncology Group, Instituto de Investigación Sanitaria Aragón (IIS Aragón), 50009 Zaragoza, Spain; rgonzalez@iisaragon.es; 2Medical Oncology Unit, Hospital Universitario Miguel Servet, 50009 Zaragoza, Spain; rpazo@salud.aragon.es; 3Digestive Radiology Unit, Hospital Universitario Miguel Servet, 50009 Zaragoza, Spain; lsarriao@salud.aragon.es; 4Biomedical Application of Radioisotopes and Pharmacokinetics Unit, Centro de Investigaciones Energéticas, Medioambientales y Tecnológicas (CIEMAT), 28040 Madrid, Spain; 5Fundación Aragonesa para la Investigación y el Desarrollo (ARAID), 50018 Zaragoza, Spain

**Keywords:** PDAC, pancreatic cancer, diagnostic imaging, immuno-PET

## Abstract

Diagnosis of pancreatic ductal adenocarcinoma (PDAC) by current imaging techniques is useful and widely used in the clinic but presents several limitations and challenges, especially in small lesions that frequently cause radiological tumors infra-staging, false-positive diagnosis of metastatic tumor recurrence, and common occult micro-metastatic disease. The revolution in cancer multi-“omics” and bioinformatics has uncovered clinically relevant alterations in PDAC that still need to be integrated into patients’ clinical management, urging the development of non-invasive imaging techniques against principal biomarkers to assess and incorporate this information into the clinical practice. “Immuno-PET” merges the high target selectivity and specificity of antibodies and engineered fragments toward a given tumor cell surface marker with the high spatial resolution, sensitivity, and quantitative capabilities of positron emission tomography (PET) imaging techniques. In this review, we detail and provide examples of the clinical limitations of current imaging techniques for diagnosing PDAC. Furthermore, we define the different components of immuno-PET and summarize the existing applications of this technique in PDAC. The development of novel immuno-PET methods will make it possible to conduct the non-invasive diagnosis and monitoring of patients over time using in vivo, integrated, quantifiable, 3D, whole body immunohistochemistry working like a “virtual biopsy”.

## 1. Introduction

Despite multiple diagnostic and therapeutic advances, pancreatic ductal adenocarcinoma (PDAC) presents a high mortality rate, representing the fourth cause of cancer death in developing countries [1,2]. This lethality can be associated with a late diagnosis, caused by the absence of symptoms at an early stage of the disease. Most cases of PDAC are located in the head of the pancreas (70%), followed in frequency by the uncinate process (18.66%), body (10–20%), and tail (5–10%) [3,4]. At present, complete surgical resection is the only potentially curative treatment for these tumors. However, only the initial stages benefit from surgery, representing only 10–15% of patients [5,6,7]. In only 10% of cases, the lesion is limited to the pancreatic gland and surrounded by normal pancreatic tissue [5,8]. At the time of diagnosis, 40–50% of cases present distant metastases, and approximately 40% of patients present signs of locally advanced disease; therefore, surgery in these cases is not indicated.

Several imaging techniques for PDAC diagnosis are available, including computed tomography (CT), magnetic resonance imaging (MRI), or endoscopic ultrasound (EUS) [9,10]. While they are widely used in the clinic and are very useful for the diagnosis of PDAC, they present several limitations.

Unlike other neoplastic processes (breast, colon, prostate…) there are no effective diagnostic screening methods for PDAC. Furthermore, due to the absolute low risk of developing this disease, population screening is not indicated. Only, in those groups [11] considered to be at-risk population, monitoring by pancreatic MRI or Cholangio-MRI, and EUS is indicated to detect small precursor lesions, such as cystic neoplasms. In these cases, CT would provide a suboptimal degree of lesion detection, compared to EUS and MRI, besides being a source of radiation [11]. Additionally, the probability of detecting lesions using these techniques is low, no more than 20% [12,13].

The development of “omics” has identified potentially relevant alterations in PDAC that still need to be integrated into the clinical management of PDAC patients. This is due, in part, to the deficiency of non-invasive imaging biomarkers [14]. “Immunotargeted imaging” represents a novel, innovative, and attractive option that combines the target specificity and selectivity of antibodies, and their variants, toward a biomarker with given imaging technique capabilities.

In this review, we describe and analyze the current diagnostic limitations of the most widely used imaging techniques in the clinic for the diagnosis of PDAC and describe the current status and promises of the immuno-positron emission tomography (PET) imaging for this devastating tumor.

## 2. Current Status of PDAC Imaging

When PDAC is suspected, diagnostic imaging techniques have two main purposes: evaluating the relationship of the tumor with the mesenteric and portal vessels and the detection of metastatic disease [15,16].

Nowadays, there are various effective imaging techniques in diagnosing and staging both local and distant pancreatic lesions. The most widely used are CT, MRI, or EUS. CT is the initial technique of choice, as recommended in the various international consensus guidelines for suspected PDAC [16,17,18,19,20].

### 2.1. Computed Tomography

Multidetector CT (MDCT) is the diagnostic technique of choice for suspected pancreatic neoplasia, as stated in numerous international guidelines [16,17,18,19,20]. The standard CT protocol in the diagnosis of PDAC includes the acquisition of 0.5–1 mm thick images with two phases: parenchymal phase (40–50 s) and venous phase (56–70 s). The parenchymal phase achieves the maximum enhancement of the pancreatic tissue that allows better detection of pancreatic lesions and assesses their relationship with adjacent arterial structures (mainly superior mesenteric artery and celiac trunk). The venous phase makes it possible to determine its relationship with the porto-mesenteric axis and have better detection of liver or peritoneal metastases. CT with multidetector technology allows image acquisition with the possibility of reconstruction in different planes (axial, coronal, and sagittal) in both phases, which improves the assessment of the relationship of the tumor with the adjacent structures [12]. It is essential to acquire images with an adequate technique and have them evaluated by experienced radiologists, thus demonstrating a significant improvement in pre-surgical staging.

In most cases (81%), PDAC is found as a hypoattenuating lesion in both arterial and venous phases because it consists of hypovascular lesions with a large desmoplastic component. On the contrary, in 11–14% of the cases, the lesions may appear isodense compared with the rest of the parenchyma, mainly in those smaller than 2.5 cm, and may occasionally be missed [15]. Other suspicious signs can be predictive of neoplasia, such as ductal dilation (sensitivity 50% and specificity 78%), hypoattenuation (sensitivity 75% and specificity 84%), ductal interruption (sensitivity 45% and specificity 82%), distal pancreatic atrophy (sensitivity 45% and specificity 96%), alteration of the pancreatic contour (sensitivity 15% and specificity 92%), and dilation of the common bile duct (sensitivity 5% and specificity 92%) [21]. In the diagnosis of PDAC, CT presents sensitivity and specificity of 89% and 90%, respectively, similar to MRI, according to various meta-analyses [22]. With the implementation of the multidetector technique, a significant improvement in sensitivity has been shown, up to 96% [23]. However, this sensitivity is reduced in small lesions, up to 65–75% (example in Figure 1A,B).

CT also allows the detection of distant metastases, liver, lung, or peritoneal infiltration. In the latter case, the existence of ascites, irregular peritoneal thickening, nodular thickening of the intestinal wall, or omental infiltration are suspicious signs [12].

In the diagnosis and staging of pancreatic cancer, PET/CT (Positron Emission Tomography/Computed Tomography) has a limited value. To date, few PET agents have been developed, with 2-[18F]fluoro-2-deoxy-D-glucose ([^18^F]FDG) being the most widely used in clinical radiopharmaceutical practice, accounting for more than 90% of worldwide studies with PET [24]. However, [^18^F]FDG is ineffective for non-invasive diagnostic imaging of PDAC. [^18^F]FDG uptake is observed at sites of metabolic activity different from tumors and metastases, such as inflammation and infection sites, and in lymphoid tissues, muscle, and brown fat (example in Figure 1C,D). PET with [^18^F]FDG is also less effective in detecting desmoplastic and hypocellular tumors and lesions with low metabolic activity [25]. In the largest, prospective, multicenter study published to date, the addition of [^18^F]FDG has been considered to be used as a standard diagnostic workup of PDAC [25]; this technique has proven to correctly change the stage (up-staging) of 10% of cases, avoiding pointless resection in 20% of patients scheduled for surgery and influencing the planned management in about 50% [25]. Nevertheless, the use of PET in PDAC diagnosis is not routinely recommended, currently restricted to clarifying equivocal findings on CT [26]. It is more useful in monitoring recurrence and response to adjuvant treatment [27,28].

### 2.2. Surgery Strategy

In the absence of distant metastasis, it is necessary to determine the resectability of the lesion based on the agreed criteria (Table 1) regarding the relationship of the tumor with arterial (celiac trunk, hepatic artery, and superior mesenteric artery) or venous (portal vein and porto-mesenteric axis) vascular structures. Vascular invasion is relatively frequent (21–64%), venous invasion being more frequent than arterial invasion. Although there is no consensus on the criteria that determine vascular infiltration, a high probability of vascular infiltration is considered when there is a direct contact between the tumor and the vessel surface, greater than 180° of its circumference (unresectable, locally advanced pancreatic cancer -LAPC-). On the contrary, it is considered of low probability when such contact is less than 180° (borderline resectable pancreatic cancer- BLPC-). There are highly specific infiltration signs, such as contour irregularity, deformity, decreased caliber, or occupancy of the vascular lumen, although all of them are of low sensitivity [29]. However, of the tumors considered resectable by CT, various meta-analyses have shown that around 40% of the patients present an under-staging when compared to the findings in the surgical act according to these criteria. This can be due to local tumor invasion, lymph node metastasis, or the presence of small hepatic or peritoneal metastases not visible by diagnostic imaging techniques [30] (example in Figure 2). Thus, the precision in the assessment of vascular invasion by CT has a sensitivity of 60% and a specificity of 94% [31]. Applying the agreed radiological criteria for resectability (Table 1), CT has a sensitivity in detecting unresectable tumors ranging 52–91% with specificity ranging 92–100% [32,33]. On the other hand, MDCT presents a positive predictive value in determining resectability of 85–89%. However, if the histological evaluation of negative margin (R0) is considered, the results are reduced to 73% [34]. Nonetheless, tumor resectability and its contingency on the vascular invasion is an evolving concept in the era of neoadjuvant therapy. The low number of potentially resectable PDAC are usually treated with upfront surgical resection followed by adjuvant chemotherapy, whereas BLPC and LAPC are typically treated initially with chemotherapy or chemoradiotherapy; subsequently, complete surgical resection can be successfully performed in up to 50% and 20% of those patients, respectively [35,36,37,38].

### 2.3. Magnetic Resonance Imaging

MRI is similar to CT in diagnosis and staging with a sensitivity and specificity of 89% in both cases [22]. However, in lesions smaller than 2 cm, MRI is superior to CT, detecting up to 79% of the lesions not initially visible with CT [16,39]. Except in cases of nephropathy or allergy to iodinated contrast agents, MRI is considered a second-line technique. MRI is sometimes used when there is a high clinical suspicion of neoplasia with negative CT [12].

### 2.4. PET Hybrid Imaging

[^18^F]FDG PET/CT and [^18^F]FDG PET/MRI have been proposed to be considered one of the routine imaging examinations used in the staging workup of PDAC. [^18^F]FDG PET/CT and [^18^F]FDG PET/MRI showed high specificity for detecting lymph node metastasis and high sensitivity and specificity for identifying distant metastasis in PDAC patients in a meta-analysis [40]. Furthermore, [^18^F]FDG PET/CT and [^18^F]FDG PET/MRI had a significant impact on the clinical management of PDAC, showing a pooled proportion of 19% of patients who underwent management changes following imaging [28]. However, due to the small number and heterogeneity of the included studies this meta-analysis requires further prospective studies with larger populations to confirm and expand these results [28].

### 2.5. Chemotherapy and Radiotherapy Response

Currently, the assessment of the response to neoadjuvant treatment in patients with a potentially resectable neoplasia or those with a locally advanced tumor using diagnostic imaging techniques is difficult. The radiological criteria, both in CT and MRI, on which the response is based, are morphological, assessing the size change of the lesion according to the Response Evaluation Criteria In Solid Tumors RECIST 1.1 criteria [41]. However, this method has limitations. These criteria are unreliable to determine treatment response and may underestimate the response or overestimate non-resectability (after neoadjuvant treatment). This is due to the progressive replacement of the tumor by fibrous tissue that can encompass the residual tumor cells without significantly modifying their size [42]. Some signs may suggest a response without changes in size, such as decreased density or the better delimitation of the tumor contour [43]. Radiologically stable disease (SD), according to RECIST1.1 criteria, accounts for most of the responses to neoadjuvant therapy. In the absence of progressive disease, the current recommendation is that patients, even just showing SD, should undergo surgical exploration [44]. Additionally, in patients with elevated carbohydrate antigen 19-9 (CA 19-9, a cell surface glycoprotein complex most commonly associated with PDAC), basal plasma levels [45], a reduction of >50% of this biomarker concentration after neoadjuvant therapy may help to select patients who will probably benefit from tumor resection [46].

Some data show that [^18^F]FDG could help to monitor response to neoadjuvant treatment, namely identifying those BLPC/LAPC patients with a complete metabolic response as the most likely candidates to achieve a complete surgical resection [47]. Notwithstanding, PET–CT has not been proven effective in evaluating response in this setting in properly prospective designed clinical trials.

After surgical resection of PDAC, recurrences are very frequent. [^18^F]FDG has a lower accuracy than CT in detecting liver metastases. It may still play a complementary role when monitoring postoperative bed, peritoneum recurrences [48], particularly when disease recurrence is suspected despite negative or equivocal CT findings (Figure 3A–D). Nonetheless, the level of evidence supporting the use of [^18^F]FDG surveillance after PDAC resection is, once more, very weak [49].

Finally, in the metastatic setting, quality of data regarding [^18^F]FDG utility is also scarce. In the phase III MPACT trial, comparing the combination of nab-paclitaxel plus gemcitabine versus gemcitabine monotherapy as first-line therapy in metastatic PDAC patients, an [^18^F]FDG metabolic response (defined as a reduction standardized uptake value -SUV- >25%) after 8 weeks on treatment resulted, in both cohorts, in a significantly higher overall response rate and more prolonged overall survival than in non-metabolic responders [50]. The rate of metabolic response (PET) was substantially higher than the RECIST response (CT), suggesting that the first one may be the more sensitive tumor response measure (Figure 3E,F). Besides, the > 3 months longer median overall survival observed for patients with a metabolic response only than for patients who did not experience a response by either measure suggested that [^18^F]FDG response may be more sensitive to treatment benefit, even in the absence of tumor response by RECIST [47]. These findings would require validation in future studies.

### 2.6. Immunotherapy and Radiomics

The introduction of immunotherapy has also changed the interpretation of response to treatment using diagnostic imaging techniques. In general, four response patterns can be found. The first corresponds to the traditional decrease in size without the existence of new lesions. The second presents a long period of radiological stability throughout the treatment, with a late reduction in the lesion, correlated with the immune system’s response time. The third type presents an initial increase in size, while the immune response system is activated, with a subsequent size reduction. The fourth pattern would correspond to the appearance of new lesions after completing the treatment preceding the tumor’s reduction, probably corresponding to micrometastases that were not initially visible and may initially increase in size, which makes them visible [48]. These last two patterns are what is called pseudo-progression [51]. Although there is no experience in the behavior after treatment with immunotherapy in pancreatic carcinomas, their behavior may follow these criteria.

The introduction of artificial intelligence methods (radiomics) in analyzing the data obtained through imaging techniques can offer an advantage in early detection programs. The application of this radiomics can improve the monitoring of treatment response by detecting small morphological variations in imaging techniques. However, it currently has significant limitations, mainly in small lesions, in which it is not possible to accurately assess various characteristics, such as density, contours, etc. [52].

## 3. Novel Non-Invasive Immunotargeted Imaging Methods for PDAC

The revolution in cancer genomics has uncovered clinically relevant alterations that have yet to be integrated into patients’ clinical management, in part due to the lack of non-invasive imaging biomarkers [14]. An innovative and attractive option is termed “immunotargeted imaging”. This approach combines the target selectivity and specificity of antibodies and engineered fragments toward a given tumor cell surface marker with the capabilities of a given imaging technique.

### 3.1. Immunotargeted Imaging Features

To develop immunotargeted imaging, three features must be taken into account (Figure 4):

#### 3.1.1. Selection of a Specific Molecular Target for Imaging

A suitable epitope for immunotargeted imaging is required to fulfill certain criteria: (1) the target needs to be exposed on the extracellular surface of the plasma membrane or to extracellular components for an easy recognition, (2) it needs to be highly expressed in the tumor, and (3) it is required to have low/no expression in normal tissue.

An ideal biomarker should predict prognosis and therapeutic response. A valuable candidate target for immunotargeted imaging should help to identify any putative association between the candidate target with therapeutical effects in PDAC.

The increased number of massive PDAC-specific databases containing multi-omics data (genome, epigenome, transcriptome, proteome, and metabolome among others) and clinical data, together with the development of novel bioinformatics tools, allow the identification of novel biomarkers that could be exploited to develop immunotargeted imaging probes [53].

#### 3.1.2. Selection of the Optimally Engineered Antibodies for Imaging Applications

The exquisite specificity of antibodies enables the targeted imaging of single biomarkers and cell types. Intact antibodies function well as therapeutics due to their long serum half-life (from days up to 3 weeks), which increases the exposure of the affected tissues to the antibody [54,55,56]. However, the long half-life of intact antibodies limits their use as imaging agents since several days are required for the blood and background clearance necessary to achieve an acceptable signal-to-noise ratio [57]. Antibodies can also be engineered as fragments with different pharmacokinetics without compromising their antigen specificity and affinity (Figure 4). The clearance of antibody fragments can be influenced by their size, charge, and hydrophobicity/hydrophilicity, as well as any fused or conjugated moieties [58]. Reduction of the overall fragment size results in accelerated blood clearance, and the removal of the antibody Fc region is common for clinical use as it reduces the molecular weight below the threshold for renal clearance (~60 kDa) and eliminates unnecessary Fc-mediated functions [59]. Other advances in protein engineering have allowed for reduced antibody size without compromising their antigen specificity and affinity. One of the most common formats is called the single-chain variable fragment (scFv, ~30 kDa), which covalently binds a light chain variable domain (VL) with a heavy chain variable domain (VH) through a flexible peptide. Crucially, this linker can be modified to allow cell permeability and blood-brain barrier (BBB) penetration. Other engineered antibody fragments with optimal pharmacokinetic properties for targeted imaging include diabodies (dimers of scFv connected by a linker that is too short to allow pairing between the two domains on the same chain), minibodies (scFv fragment fusions with the antibody constant domain), and nanobodies (~15 kDa heavy chains derived from those found in *Camelidae* species) [60]. Each of these fragments retains the high affinity and specificity of the parental antibody while exhibiting optimal blood-clearance properties. Notably, compared to full-length antibodies, each of these small antibody fragments can be advantageous for studying brain metastases in PDAC as they can more efficiently cross the BBB, a membrane that prevents most large drug molecules from entering the CNS [54,61].

Recently, even smaller sized peptides based on *Staphylococcus aureus* protein A (affibodies, 58 amino acids, ~7 kDa) have been generated and demonstrate affinity to other molecules [62,63]. The small sizes of these affibodies (and nanobodies) enable them to bind to epitopes that intact antibody and Fv-based fragments cannot access [64]. They are also particularly suitable for applications in which extremely rapid clearance is desired. It should be noted that interspecies immunogenicity of antibodies and their derivatives can be restrictive for their clinical use. The “humanization” of non-human antibodies and fragment derivatives by modifying their protein sequences to be more similar to that of human antibodies is commonly undertaken to prevent unwanted immune responses [65].

#### 3.1.3. Selection of the Suitable Modality-Specific Imaging Agent

The choice of imaging modality and tracer depends on several factors, including the required sensitivity, resolution, and whether quantitation and multiplexing are possible [68,69]. For clinical applications, nuclear medicine-based imaging modalities (i.e., single-photon emission computed tomography [SPECT] and PET) that detect gamma rays emitted by a radiotracer confer high sensitivity and are quantifiable. Optical imaging provides an important alternative for molecularly targeted imaging since it avoids the use of ionizing radiation. However, optical imaging with fluorescently labeled probes, including those using antibodies, has been limited in application due to tissue scattering, photons’ absorption in the visible range (400–800 nm), and background signal from autofluorescence [70]. Antibodies have also been employed to impart specificity to MRI and ultrasound imaging, although thus far, applications have been limited to preclinical settings. Targeted MRI requires the conjugation of contrast agents such as gadolinium (Gd)-complexes or superparamagnetic iron oxide (SPIO) nanoparticles to a specific probe [65]. Ultrasonic nanobubbles coupled with specific antibodies for targeted ultrasound echography have been developed [71]. Immuno-imaging with computed tomography (CT) is more problematic. Still, the use of antibody-labeled gold nanoparticles and antibody-conjugated liposomes (immunoliposomes) containing CT contrast agents potentially allows immuno-CT imaging. Of note, antibodies are particularly suited to be developed as multi-modal tracers, or as combination therapeutic/diagnostic (“theranostic”) agents, due to the ability to conjugate them with a variety of cargoes with minimal or manageable impact on pharmacokinetics, biodistribution, and clearance. A recent example is a dual-labeled antibody with both PET and optical imaging functions that can be used for imaging prostate cancer both pre-and intra-surgery and is being adapted to other cancer types [72,73,74].

An extensive revision of other imaging approaches with antibodies can be found in England et al. [75].

### 3.2. The Current Status of Immuno-PET in PDAC

We have selected to review here the status and promise of the immuno-PET imaging techniques in PDAC. By merging the high target specificity of antibodies with the high spatial sensitivity, quantitative capabilities, and resolution of PET, it is possible to conduct the non-invasive diagnosis and monitoring of patients over time using in vivo, integrated, quantifiable, 3D, full-body immunohistochemistry (Figure 4).

#### Selection of a Suitable Radionuclide for Immuno-PET

For immuno-PET, it is important to match the physical half-life of the positron-emitting radionuclide with the biological half-life of the antibody or fragment being used. PET radionuclides with longer half-lives, such as ^89^Zr (t_1/2_ = 78.4 h, ß^+^mean 395.5 keV, ß^+^ Yield 89%) and ^124^I (t_1/2_ = 100.3 h, ß^+^mean 687 keV, ß^+^ Yield 89%), should be conjugated with intact antibodies (t_1/2_ = days to weeks) [76]. Smaller fragment derivatives can be labeled with PET isotopes characterized by intermediate half-lives, such as ^64^Cu (t_1/2_ = 12.7 h, ß^+^mean 278.2 keV, ß^+^ Yield 18%) or ^86^Y (t_1/2_ = 14.7 h, ß^+^mean 535.4 keV, ß^+^ Yield 34%), or short half-lives, such as ^18^F (t_1/2_ = 68 min, ß^+^mean 249.8 keV, ß^+^ Yield 97%), or ^68^Ga (t_1/2_ = 68 min, ß^+^mean 836 keV, ß^+^ Yield 89%) and ^44^Sc (t_1/2_ = 3.94 h, ß^+^mean 632 keV, ß^+^ Yield 94.27%).

In addition to its short half-life, a key advantage of ^68^Ga or ^44^Sc is that they can be produced from a commercially available ^68^Ge/^68^Ga or ^44^Ti/^44^Sc generator allowing their production to be cyclotron-independent, making it accessible to any PET center and at a lower economic cost [77,78,79]. With regards to ^44^Sc, as it emits prompt gamma-rays right after the positron emission, it can be distinguished from standard positron emitters like ^68^Ga or ^18^F, enabling multiplexed PET (mPET) imaging [80,81,82]. This technique allows for the accurate simultaneous non-invasive imaging of two different radiotracers with preclinical and clinical PET scanners [82].

Radionuclides can either be directly conjugated to an antibody via radiohalogenation onto random tyrosine residues, or they can be attached indirectly through a linker (like hexadentate tris(hydroxamate) siderophore desferrioxamine-B (DFO), 1,4,7,10-tetraazacyclododecane-1,4,7,10-tetraacetic acid (DOTA), 1,4,7-triazacyclononane-1,4,7-triacetic acid (NOTA)) [83] that contains a chelating group for attachment of radiometals and a reactive group reacting with ε-amino groups of lysine residues and/or N-terminus of a protein [78,84]. Bioconjugation of the linker to the antibody can be alternatively accomplished by a “biorthogonal reaction” [85], which itself is an extension of “click chemistry.” Bioorthogonal reactions must (1) produce a chemically and biologically inert linkage/product via a reaction that displays high selectivity between the two coupling partners (i.e., azide and alkyne), (2) be kinetically fast, and (3) be biocompatible in terms of operating at physiological pH, temperature, and in a physiologically relevant solvent milieu. Some specific bioorthogonal reactions can occur in vivo, allowing for a two-step pretargeting strategy. A primed antibody or fragment, which has already been linked to one of the reaction components, can be administered before the reaction is complete [86,87]. After some time, possibly a few hours or days depending on the antibody half-life, the second component (i.e., a chelating agent containing the radionuclide) of the reaction can be administered. Two-step pretargeting allows for smaller doses of radioactive material to be used and provides faster clearance, reducing patients’ exposure to radioactivity and ensuring a better signal-to-noise ratio. Furthermore, this strategy can be used for the labeling of different tracers (MRI-tracers such as (Gd)-complexes or SPIO nanoparticles) to the same pretargeted molecule to allow for multi-modal and/or multifunctional imaging [88,89].

As previously described, [^18^F]FDG PET imaging present multiple limitations for PDAC diagnosis [90]. PET imaging using radiolabeled monoclonal antibodies (Immuno-PET) provides a non-invasive and whole-body visualization of in vivo antibody biodistribution. Immuno-PET, which exquisitely fuses the extraordinary targeting specificity of mAb and the superior sensitivity and resolution of PET, is a paradigm shift for molecular imaging modalities. The clinical application of immuno-PET imaging has increased the understanding of tumor heterogeneity and refined clinical disease management. Besides monoclonal antibodies, other immuno-PET probe formats, ranging from antibody-derived fragments to nanobodies, have increased interest due to their faster pharmacokinetics and enhanced imaging characteristics [91]. Nanobodies are single-domain variable regions (VHHs) derived from camelid heavy-chain-only antibodies and have significant advantages over other antibody formats for in vivo diagnostic imaging and targeted delivery. Their small size (~15 kDa) allows deeper tissue penetration and faster renal clearance than larger antibody reagents. Nanobodies are reported to be poorly immunogenic and highly stable [92,93,94].

Immuno-PET applications require simple, fast, and specific radiolabeling of antibody-based probes under mild conditions. Optimal immuno-PET imaging is attributed to a highly specific tumor uptake and low background retention. Toward this end, it is essential for a tracer to specifically saturate its target as fast as possible, with the unbound tracer cleared out rapidly from the blood circulation.

The successful development of immuno-PET probes is highly dependent on the choice of tumor-targeting antibodies and derivatives, radionuclides, bifunctional chelators, and conjugation strategies. mAb radiolabeling pioneered with SPECT radionuclides (^131^I,^123^I, ^111^In, ^99m^Tc). However, over the years, the nuclear medicine community’s interest has shifted towards PET radionuclides (^89^Zr, ^64^Cu, ^124^I, ^86^Y, ^68^Ga, ^18^F). Optimized nuclear reactions and higher purity make these radionuclides more readily available. Furthermore, PET scanners allow the acquisition of higher resolution images and present higher sensitivity. Altogether, PET imaging allows a more-accurate image quantification. However, PET radionuclides present a higher production cost and higher radiation burdens due to these radionuclides’ higher photon energies. This increased exposure was often balanced in practice since SPECT tracers required higher injected activities due to the lower detector sensitivity [95,96]. It is expected that the progressive increase of novel detectors and improved system designs will increase sensitivity, enabling the administration of lower administered activities facilitating the immuno-PET/SPECT use in the near future [97].

Immuno-PET/SPECT permits detecting functional tumor biomarker changes allowing an earlier diagnosis of PDAC and monitoring of patients. Of importance, biomarker changes can occur earlier than a reduction in tumor size, as it usually represents late treatment effects.

Notably, immuno-PET allows quantification of biomarkers in a non-invasive manner in the whole body. Current quantification of biomarkers in PDAC requires a biopsy analysis by immunohistochemistry (IHC) and molecular biology assays. A single biopsy usually does not capture the tumor heterogeneity and requires repetitive biopsies and histopathological confirmation to monitor treatment response. These hurdles represent a clinical challenge and a risk for the patients. On the other hand, immuno-PET provides a whole body, non-invasive, quantitative, and longitudinal evaluation of tumor target expression and distribution.

The analysis of tumors detecting biomarkers in the blood is beginning to transform cancer diagnosis. Immuno-PET will be complementary to liquid biopsies. While liquid biopsies can also identify patients with treatment failure or relapse, subsequent imaging is frequently required to localize and characterize the disease and guide subsequent treatment decisions [97,98,99,100].

Several targets are functionally important in PDAC since they have clinical potential as a prognostic marker. Moreover, they could be used as a target for the delivery of agents for its detection. As shown in Table 2, membrane proteins that are overexpressed on tumor or tumor-associated cells have been potentially suitable for tumor-targeted imaging; other components of the tumor microenvironment, such as extracellular matrix proteins, have also been promising candidates for the development of diagnostic approaches in PDAC. 

With the rise of immunotherapy in recent years, PET imaging of immune checkpoint inhibitors (ICIs) may serve as a robust biomarker to predict and monitor responses to ICIs, complementing the existing immunohistochemical techniques [91,123]; it has been described that PET imaging using antibodies against the programmed cell death receptor 1 (PD-1)/programmed cell death ligand 1 (PD-L1) pathway can be a useful method for evaluating PD-L1 expression in orthotopic pancreatic cancer models [124]. To date, most of the PET imaging probes have been designed to target PDAC tumors in preclinical models (Figure 5), and only one study has been conducted with an [^89^Zr]Zr-labeled human monoclonal antibody in patients with pancreatic cancer or other CA19-9 positive malignancies [125].

## 4. Discussion

Diagnosis of PDAC by current imaging techniques (CT, MRI, PET, and EUS) is useful and widely used in the clinic but presents several limitations, especially in small lesions. It is not possible to accurately assess various characteristics, such as density, contours, or others. These limitations frequently cause radiologically infra-staging of tumors, false-positive diagnosis of metastatic tumor recurrence, and common occult micro-metastatic disease. Local inflammatory, fibrotic changes at the surgical site and treatment response evaluation are diagnostic challenges for radiologists (Figure 1, Figure 2 and Figure 3). Novel treatments, such as immunotherapies, have also changed treatment response interpretation using diagnostic imaging techniques [126]. The use of artificial intelligence will strengthen these techniques but still present significant restraints.

The development of multi-“omics” and bioinformatics have uncovered clinically relevant alterations in PDAC that still need to be integrated into the clinical management of patients. One urgent need is the development of non-invasive imaging biomarkers to assess and integrate this information into the management of patients [14].

“Immunotargeted imaging” represents an innovative and attractive option for the diagnosis of PDAC. It combines the target specificity and selectivity of antibodies and variants toward a biomarker with a given imaging technique’s capabilities. Of importance, functional tumor biomarker expression changes can occur earlier than changes in the lesion size as assessed on morphological imaging [127].

Immuno-PET merges the high target selectivity and specificity of antibodies and engineered fragments toward a given tumor cell surface marker with the high spatial resolution, sensitivity, and quantitative capabilities of PET imaging techniques. The development of novel immuno-PET methods will make it possible to conduct the non-invasive diagnosis and monitoring of patients over time using in vivo, integrated, quantifiable, 3D, whole body IHC [128]. Current quantification of biomarkers in PDAC requires biopsies and anatomopathological and molecular biology analysis that might not capture the complete tumor heterogeneity. Importantly, immuno-PET allows quantification of biomarkers in a non-invasive and longitudinal manner in the whole body, like a “virtual biopsy” [129].

Several targets are functionally important in PDAC and might have clinical potential as prognostic biomarkers to be used in immuno-PET. Proteins and molecules present at the plasmatic membrane that are overexpressed on tumor or its microenvironment are potentially suitable for tumor-targeted imaging. Other components of the tumor microenvironment, such as extracellular matrix proteins, arise as promising candidates for the development of immuno-PET probes for diagnosis and monitoring of PDAC patients [130]. To date, most of the PET imaging probes have been designed to target PDAC tumors in preclinical models (Table 2). Only one study has been conducted with an ^89^Zr-labeled human mAb in patients with pancreatic cancer or other CA19-9 positive malignancies [125]. With the rise of immunotherapies over the past decade, PET imaging of immune checkpoint inhibitors may serve as a robust biomarker to predict and monitor responses in PDAC [91,123].

For a successful immuno-PET probe, it is crucial to match the positron-emitting radionuclide’s physical half-life with the biological half-life of the antibody or fragment being used. The slow clearance of intact antibodies limits their use as imaging agents since several days are required to achieve a satisfactory signal-to-noise ratio [57] and the patient would be exposed to radioactivity for an extended period [131]. To solve this issue, some biorthogonal reactions that can occur in vivo could be used. This approach would allow a two-step pretargeting approach, reducing the doses of radioactive material to be used and providing a faster clearance, thereby reducing patients’ exposure to radioactivity and improving the signal-to-noise ratio [85,86,87].

Although antibody-based therapies are widely used in patients’ clinical care, antibodies present multiple limitations, including their large size and low penetration in solid tissues [92]. Nanobodies are emerging as an alternative. This novel and unique class of antigen-binding fragments are derived from heavy-chain-only antibodies naturally present in the serum of *Camelidae* [92,93,94]. They exhibit advantageous properties such as small size, high stability, water-solubility, strong antigen-binding affinity, and natural origin make them suitable for development into the next-generation of biodrugs [92].

Recently, caplacizumab (ALX-0681), a bivalent nanobody [132] for the treatment of patients suffering from thrombotic thrombocytopenic purpura, received approval from the European Medicines Agency (EMA) and the US Food and Drug Administration (FDA), giving domain antibodies in the clinic and research a boost. Nanobodies can be labeled with PET isotopes of shorter half-lives, such as ^68^Ga, which advantageously can be produced in a typical PET center generator.

Liquid biopsies and other analysis detecting tumoral biomarkers in the body fluids are transforming cancer diagnosis and patient monitoring. Novel quantitative, specific, and sensitive imaging methods, such as immuno-PET, will be required to localize the lesion and guide successive therapeutic decisions [97,98].

## Figures and Tables

**Figure 1 jcm-10-01151-f001:**
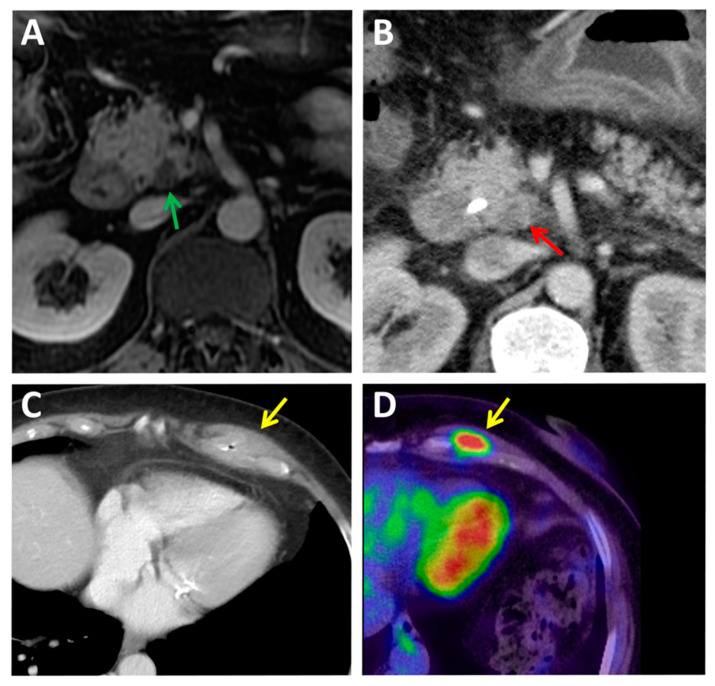
Cases of radiologically infra-staged and false-positive diagnosis of metastatic tumor recurrence by current imaging methods. (**A**,**B**) A radiologically infra-staged T2 N0 pancreatic ductal adenocarcinoma (PDAC) was found to be a locally advanced pT4 pN1 PDAC in the pathology report. (**A**) magnetic resonance imaging (MRI) showing a small cystic area in the pancreas’ uncinate process (green arrow), a bile duct stricture, no direct signs of malignancy. (**B**) A computed tomography (CT) performed after bile stent placement showed a small hypodensity (red arrow) next to the superior mesenteric artery. (**C**,**D**) A case of a false-positive diagnosis of metastatic tumor recurrence. (**C**) Some months after surgical excision of a PDAC stage pT2 pN0 M0 R0, an asymptomatic solid mass in the left costal wall (yellow arrow) was shown by a CT scan. (**D**) An 10.43 SUVmax in 2-[18F]fluoro-2-deoxy-D-glucose ([^18^F]FDG) positron emission tomography (PET)-CT was suspicious of a PDAC metastatic relapse. A tumor core biopsy found inflammatory and fibrotic tissue but no sign of malignant cells.

**Figure 2 jcm-10-01151-f002:**
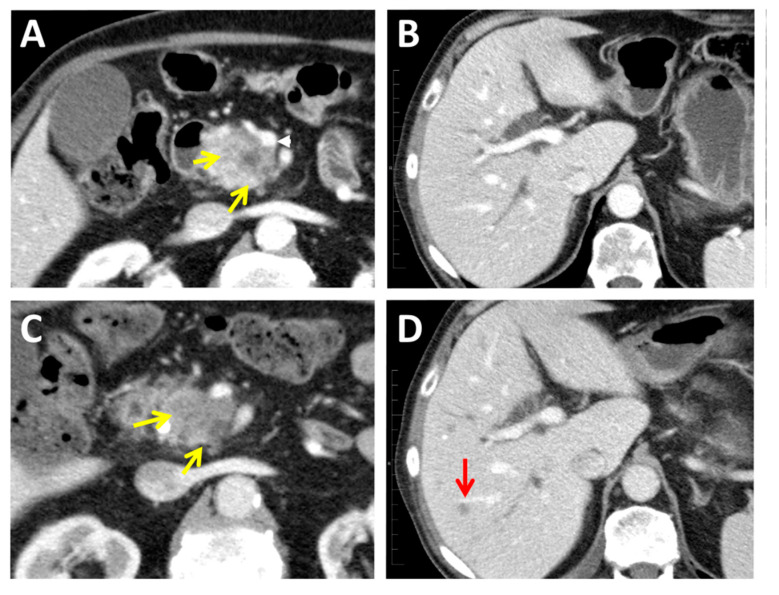
Occult micro-metastatic disease is common in PDAC, often undetected by current radiology techniques. Case. (**A**) At the time of diagnosis, a CT found 37 mm pancreatic head tumor (yellow arrows) contacting > 50% with the superior mesenteric vein (white arrow) with no liver metastasis (**B**) and classified as a resectable stage T3 N0 M0. In a CT performed just 4 weeks after the basal one, the pancreatic mass was stable (yellow arrows) (**C**), but liver metastases were found (red arrow) (**D**), the tumor re-staged as a T3 N0 M1, and the tumor surgical excision was not indicated anymore.

**Figure 3 jcm-10-01151-f003:**
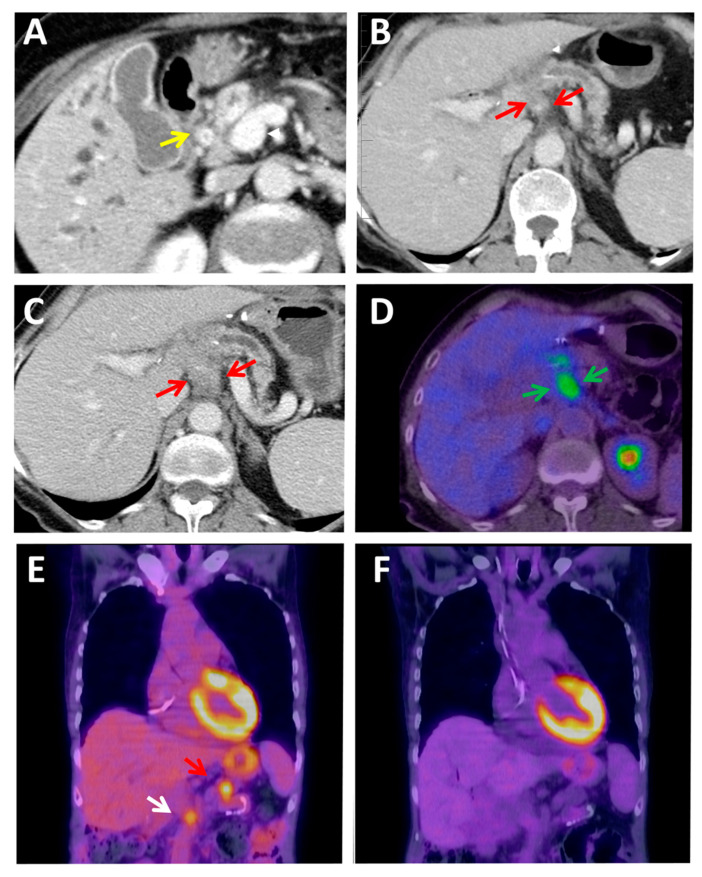
Local inflammatory and fibrotic changes at the surgical site and treatment response evaluation are diagnostics challenge for radiologists. (**A**–**D**) A case of local inflammatory and fibrotic changes at the surgical site are a diagnostic challenge for radiologists. A PDAC was suspected because of a bile duct stricture (**A**) (yellow arrow). The tumor was surgically resected; the pathology report showed a stage I cancer, pT1c pN0 M0 R1 (retroperitoneal margin was microscopically affected). Findings in a CT performed two months after surgery required a differential diagnosis between benign fibrotic tissue at the surgical site and a retroperitoneal tumor relapse (red arrows). The increasing tissue size (red arrows) in a consecutive 1-month later performed CT (**C**) and the moderately high 3.67 SUVmax found in an [^18^F]FDG PET–CT (**D**) finally made the diagnosis of relapsed PDAC, and a palliative combination chemotherapy regimen was initiated. (**E**,**F**) [^18^F]FDG PET PDAC treatment response evaluation. (**E**) PDAC relapsed at the surgery site (15 mm, 5.73 SUVmax) (red arrow) and metastatic lymph node in the superior mesenteric vein area (10 mm, 4.0 SUVmax) (white arrow). (**F**) Early complete metabolic response after two cycles of chemotherapy.

**Figure 4 jcm-10-01151-f004:**
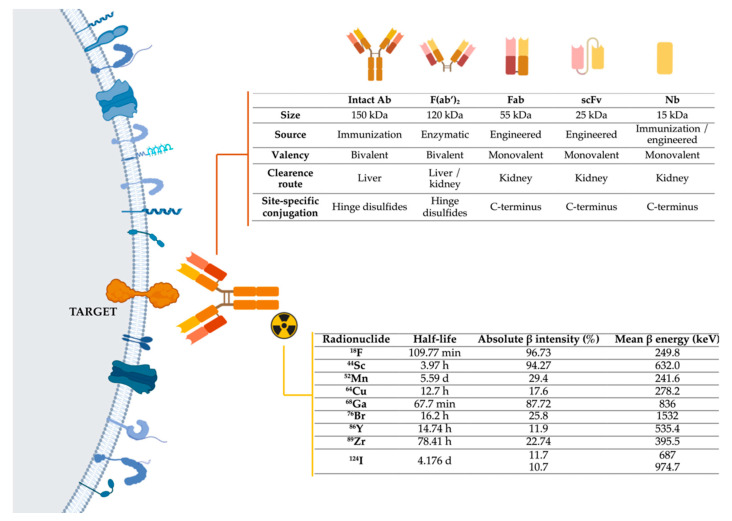
Representation of the three main components of immuno-PET techniques: target, antibodies, and radionuclides. Abbreviations: Ab-Antibody; Fab-Fragment antigen-binding; F(ab’)_2_-Fab dimer; scFv- single-chain variable fragment; Nb-Nanobody, ^18^F-fluorine; ^44^Sc-scandium; ^52^Mn-manganese; ^64^Cu-copper; ^68^Ga-gallium; ^76^Br-bromine; ^86^Y-yttrium; ^89^Zr-zirconium; ^124^I-iodine [66,67]. Image generated with BioRender.

**Figure 5 jcm-10-01151-f005:**
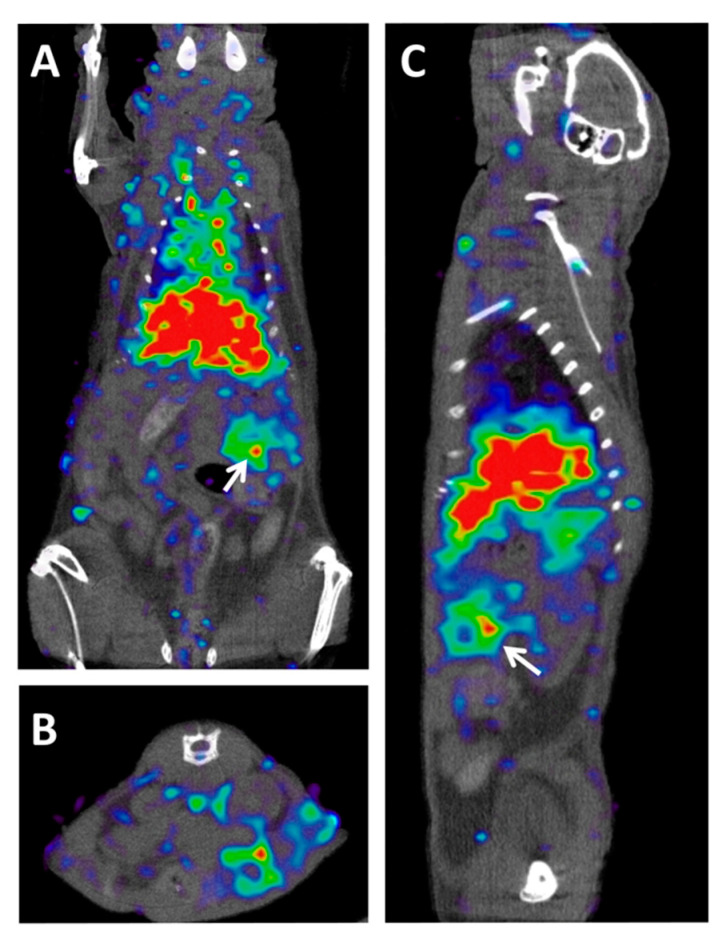
ImmunoPET–CT of MT1-MMP metalloproteinase in a preclinical model of PDAC. (**A**) Coronal, (**B**) axial, and (**C**) sagittal views of fused Immuno-PET and CT images of an orthotopic pancreatic patient-derived xenograft mouse. White arrows indicate tumor location. The imaging probe used was [^89^Zr]Zr-DFO-LEM2/15, a mAb developed against the MT1-MMP metalloproteinase [107]. Owing to the central role that this metalloproteinase plays in collagen-induced gemcitabine resistance, this probe could be used for the early prediction of resistance to gemcitabine in metastatic PDAC patients.

**Table 1 jcm-10-01151-t001:** Definition criteria of resectability by NCCN v 1.2019.

Resectability Status	Arterial	Venous
Resectable	No tumoral contact with CeT, SMA, or CHA	No tumoral contact with SMV, PV, or ≤180° without irregularity of the venous contour
Resectability borderline	Head and uncinate process: Tumoral contact with CHA without CeT extension or CHA bifurcation allowing safety and complete resection and reconstruction Tumoral contact in SMA ≤ 180°. Tumoral contact with an anatomical variant of CHA	Head and uncinate process: Tumoral contact with IVC, PV > 180° with venous contour irregularities or thrombosis, but with free proximal and distal portions that allow a suitable resection
	Body-tail: Tumoral contact with CeT ≤ 180° Tumoral contact with CeT ≥ 180° without aorta involvement, with gastroduodenal artery intact	
Unresectable	Distant metastasis (includes lymph nodes but not regionals): Head and uncinate process: Tumoral contact with AMS > 180° Tumoral contact in CeT > 180°	Head and uncinate process: SMV and PV irreconstructable by thrombosis or tumoral infiltration. Contact with the sewer system in the 1st jejunal venous branch
	Body-tail: Tumoral contact with AMS or CeT > 180° Tumoral contact with CeT and aorta	Body-tail: SMV and PV irreconstructable by thrombosis or tumoral infiltration

CeT: celiac trunk; SMA: superior mesenteric artery; CHA: common hepatic artery; SMV: superior mesenteric vein; PV: portal vein; IVC: inferior vena cava.

**Table 2 jcm-10-01151-t002:** Immuno-PET applications in PDAC.

PET Imaging Probes	Conjugation Strategy	Targets	Hallmark	Models	References
[^64^Cu]Cu-DOTA-anti-PD-1 [^64^Cu]Cu-NOTA-anti-PD-1 [^64^Cu]Cu-NOTA-anti-PD-L1	Lysine-based random	PD-1/PD-L1	Imaging of immune checkpoints	Orthotopic KRAS murine PDAC	[101]
[^89^Zr]Zr-Df-10D7 (anti-CDCP1 mAb)	Lysine-based random	CUB Domain-Containing Protein 1 (CDCP1)	CDCP1 regulates migration, invasion, and extracellular matrix degradation	Patient-derived subcutaneous and orthotopic xenografts (PDX) mice	[102]
[^64^Cu]Cu-PCTA-cetuximab	Lysine-based random	Epidermal Growth Factor Receptor (EGFR)	EGFR is overexpressed in a wide variety of cancers	Resectable orthotopic xenograft mouse model with human PC XPA-1 cells	[103]
**[^89^Zr]Zr-Df-MVT-2163** **(human HuMab-5B1 Ab)**	**Lysine-based random**	**CA19-9 (Sialyl Lewis A)**	**CA19-9 is the most commonly used serum tumor marker for PDAC**	**Patients with primary PDAC and metastases** **(Phase 1)**	[104,105]
[^64^Cu]Cu-NOTA-NJB2 (nanobody)	Sortase-Mediated Radiolabeling	Alternatively spliced EIIIB (EDB) domain of fibronectin tumor extracellular matrix and neovasculature	Fibronectin is a glycoprotein that forms a major constituent of tumor extracellular matrix and neovasculature	(K-ras^LSL.G12D/+^; p53^R172H/+^; PdxCre) KPC mouse models of PDAC	[106]
[^89^Zr]Zr-Df-LEM2/15 (anti-MM1-MMP mAb)	Lysine-based random	MT1-MMP	Metalloprotease MT1-MMP is overexpressed in many tumors and associates with tumor growth, invasion, metastasis, and poor prognosis	Subcutaneous xenograft mouse model with Capan-2 cells, and subcutaneous and orthotopic PDX mice.	[107]
[^89^Zr]Zr-Df-MEHD7945A (duligotuzumab)	Lysine-based random	EGFR and Receptor tyrosine-proteinase kinase erbB-3 (HER3)	EGFR and HER3 are highly expressed in PDAC, marking this aggressive disease with poor survival rates	Subcutaneous xenograft mouse model with AsPC-1 cells	[107]
[^124^I]-A2cDb (anti-PSCA 2B3 A2 cys-diabody) [^124^I]-A11 Mb (anti-PSCA minibody)	Direct iodination	Prostate stem cell antigen (PSCA)	PSCA is also overexpressed in pancreatic carcinoma	Subcutaneous PDX mice	[108]
[^64^Cu]Cu-NOTA-3B4 (single chain Fv)	Lysine-based random	Receptor for advanced glycation end products (RAGE)	RAGE is overexpressed in human pancreatic tumors; it is a critical promoter in the transition of premalignant epithelial precursors (PanIN) to PDAC	Balb c/nude mice bearing Panc02 tumors. No PET study, only ex vivo biodistribution.	[109]
[^89^Zr]Zr-Df-ALT-836 (anti-human TF mAb)	Lysine-based random	Tissue factor (TF)	Overexpression of TF in pancreatic cancer has been correlated with high tumor grade, the primary disease’s extent, and local and distant metastatic invasion.	Subcutaneous xenograft mouse model with BxPC-3 or PANC-1 cells	[110]
[^64^Cu]Cu-NOTA-heterodimer-ZW800 (bispecific immunoconjugate of CD105 and TF Fab′ antibody fragments)	Lysine-based random	Endoglobin (CD105) and TF	CD105 is a cell surface glycoprotein expressed on endothelial cells, and its overexpression in cancer has been linked to angiogenesis, metastasis, and cancer progression	Subcutaneous xenograft mouse model with BxPC-3 or PANC-1 cells	[111]
[^89^Zr]Zr-Df-5B1 (anti-CA19.9 mAb)	Lysine-based random	CA19-9	CA19-9 is the most commonly used serum tumor marker for PDAC	Orthotopic xenograft mouse model with CAPAN-2 cells	[112]
[^89^Zr]Zr-Df-1A2G11 (anti-IGF-1R mAb)	Lysine-based random	Insulin-like growth factor-1 receptor (IGF-1R)	IGF-1R is a transmembrane receptor of the tyrosine kinase class involved in cell growth, apoptosis, and tumor invasion in cancer	Subcutaneous xenograft mouse model with MIA PaCa-2 or BxPC-3 cells	[112,113]
[^64^Cu]Cu-DOTA-MAb159 (anti-GRP78 mAb)	Lysine-based random	Glucose-regulated protein (GRP78)	Cell-surface GRP78 expression, an immuno-globulin heavy-chain binding protein, has been detected in pancreatic cancer.	Subcutaneous xenograft mouse model with BxPC-3 cells	[114]
[^64^Cu]Cu-DOTA-11-25 (anti-Mesothelin mAb)	Lysine-based random	Mesothelin (MSLN)	MSLN is a cell differentiation-associated glycoprotein, overexpressed in various cancers, including PDAC	Subcutaneous xenograft mouse model with CFPAC-1 or BxPC-3 cells	[115]
[^89^Zr]Zr-Df-TSP-A01 (anti-transferrin receptor mAb)	Lysine-based random	Transferrin receptor (TfR)	TfR is upregulated on the cell surface of many cancer types, including pancreatic cancer	Subcutaneous xenograft mouse model with MIA PaCa-2 cells	[116]
[^89^Zr]Zr-Df-059-053 (human anti-CD147 mAb)	Lysine-based random	CD147	CD147 (so-called EMMPRIN) is a transmembrane protein of the immunoglobulin superfamily and is expressed in many types of tumors, including PDAC	Subcutaneous xenograft mouse model with MIA PaCa-2 cells	[117]
[^64^Cu]Cu-NOTA-panitumumab-F(ab′)_2_	Lysine-based random	EGFR	EGFR is overexpressed in a wide variety of cancers	Subcutaneous xenograft mouse model with PANC-1 cells, and subcutaneous and orthotopic PDX OCIP23 mice	[118]
[^89^Zr]Zr-Df-5B1 (anti-CA19.9 mAb)	Lysine-based random	CA19-9	CA19-9 is the most commonly used serum tumor marker for PDAC	Subcutaneous xenograft mouse model with BxPC3 cells	[119]
[^124^I]-A2cDb (anti-CA19.9 diabody)	Direct iodination	CA19-9	CA19-9 is the most commonly used serum tumor marker for PDAC	Subcutaneous xenograft mouse model with BxPC3 or CAPAN-2 cells	[120]
[^64^Cu]Cu-NOTA-ALT-836 (anti-human TF mAb)	Lysine-based random	Tissue factor (TF)	Overexpression of TF in pancreatic cancer has been correlated with high tumor grade, the primary disease’s extent, and local and distant metastatic invasion.	Subcutaneous xenograft mouse model with BxPC-3, PANC-1, or ASPC-1 cells	[121]
[^64^Cu]Cu-DOTA-2A3 (2A3 is an anti-CEACAM6 nanobody) [^64^Cu]Cu-DOTA-2A3-mFc (2A3 fused with a murine Fc fragment) [^64^Cu]Cu-DOTA-9A6 (anti-CEACAM6 murine mAb)	Lysine-based random	Carcinoembryonic antigen-related cell adhesion molecule 6 (CEACAM-6)	CEACAM-6 is a cell surface glycoprotein known to be highly expressed in most cancers	Subcutaneous xenograft mouse model with BxPC3 cells	[122]
[^124^I]-H310A (anti-CEA scFv-Fc)	Direct iodination	Carcinoembryonic antigen (CEA)	CEA is a GPI-linked glycoprotein overexpressed in gastrointestinal epithelial tumors, including PDAC	Subcutaneous xenograft mouse model with BxPC-3, CAPAN-1, or HPAF-II cells	[121]

Antibody-based PET imaging probes for PDAC ordered by the most recent publication date. Bioconjugation strategy has been categorized into three methods: lysine-based random, site-specific via sortase-mediated reaction, and direct iodination. Antibody-based PET imaging probes reaching clinical trials are highlighted in bold.

## Data Availability

Data is contained within the article.

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
