# Peer review of "Diagnosis of Pancreatic Ductal Adenocarcinoma by Immuno-Positron Emission Tomography"

_jcm, 2021, doi:10.3390/jcm10061151_

Round 1

Reviewer 1 Report

The presented review article of Gonzaléz-Gómez et al is a solid addition to the field, provides a wonderful overview of Immuno-Positron Emission Tomography and describes its importance for current diagnostic and partially treatment of the pancreatic ductal adenocarcinoma. The review is clearly written, has a very good structure. A comparison to previous standard methods is provided and the perspectives of the new technology are defined. There is a very nice overview of Immuno-PET studies presented in Table 2.

In summary, I am totally in favor of publication of this review.

Author Response

Dear Reviewer #1

We thank the reviewer for critical review of the manuscript entitled “Diagnosis of Pancreatic Ductal Adenocarcinoma by Immuno-Positron Emission Tomography” written by Ruth González-Gómez, Roberto A. Pazo-Cid, Luis Sarría, Miguel Ángel Morcillo, and myself for intended publication in the special issue “Pancreatic Cancer: Challenges and Breakthroughs” of the prestigious Journal of Clinical Medicine.

We appreciate his comments and celebrate and his/her positive reply.

We always intended to provide an overview of the importance of imaging for the current diagnostic of the pancreatic ductal adenocarcinoma, comparing previous standard methods with the novel perspectives of the immuno-PET.

We acknowledge that Table 2 could provide an overview of this field of imaging for PDAC.

Sincerely,

Alberto J. Schuhmacher

Reviewer 2 Report

JCM-1112714 review manuscript well provides comprehensive and prospective view of diagnosis of pancreatic ductal adenocarcinoma by immunoPET. Immune-PET applications in PDAC are well summarized as in Table 2 which will be very informative for readers. No major correction is required. However, the manuscript would benefit from additional refinement of several references and several typographical errors noted below.

Minor comments

  • Conjugation strategy in Table 2 can be categorized as three conjugation methods such as A. lysine-based random, B. Direct iodination, and C. Sortase-Mediated Radiolabeling, which might be denoted in table caption.

* line 187 and 191. [[18F] à [18F]

  • Table 1, 180º in second column and second row should be italic.
  • line 315 and 772, reference 58 should be corrected (DOI: 1158/1078-0432.CCR- 14-1452)
  • line 324, space between 15 and kDa.
  • line 389-395, inconsistency of space with time
  • line 346 and 395, 46Sc should be 44Sc
  • line 401, reference 70 by Cal-Conzalez, J. et al introduces a concept of mPET and several radionuclides, but 44Sc is not included. One of below publications or both introducing 44Sc could be additionally

Rosar, F. et al. Image quality analysis of 44Sc on two preclinical PET scanners: a comparison to 68Ga. EJNMMI Phys. 7, (2020). https://doi.org/10.1186/s40658- 020-0286-3

Rosar, F. et al. Impact of prompt gamma emission of 44Sc on quantification in preclinical and clinical PET systems. Appl. Radiat. Isot. 170, 1–8 (2021). https://doi.org/10.1016/j.apradiso.2021.109599

  • Table 2, 4th row and model column, please check bold

* line 503 and 618, 89Zr- à [89Zr]Zr-

  • line 589, please check reference 2.

* line 728,124I à 124I

  • line 804-807, reference

Author Response

Dear Reviewer #2

We thank the reviewer for critical review of the manuscript entitled “Diagnosis of Pancreatic Ductal Adenocarcinoma by Immuno-Positron Emission Tomography” written by Ruth González-Gómez, Roberto A. Pazo-Cid, Luis Sarría, Miguel Ángel Morcillo, and myself for intended publication in the special issue “Pancreatic Cancer: Challenges and Breakthroughs” of the prestigious Journal of Clinical Medicine.

We appreciate his/her comments that positively improve the manuscript.

We apologize for the various typographical errors. We acknowledge the reviewer's contributions.

We answer one by one the points

Point 1. Conjugation strategy in Table 2 can be categorized as three conjugation methods such as A. lysine-based random, B. Direct iodination, and C. Sortase-Mediated Radiolabeling, which might be denoted in table caption.

Response: We agree that certainly this clarifies the table. We have included the following sentence in the Table caption (line: “ Bioconjugation strategy has been categorized into three methods: lysine-based random, site-specific via sortase-mediated reaction, and direct iodination.” Line 481-482. We could rearrange the table according to the conjugation strategy if the reviewer considers it necessary.

Point 2.* line 187 and 191. [[18F] à [18F]

Response: We thank reviewer #2 for the observations; we have corrected them. The change is highlighted in the main text using track-changes.

Point 3. Table 1, 180º in second column and second row should be italic.

Response: We thank reviewer #2 for the observations; we have corrected them. The changes are highlighted in the main text using track-changes.

Point 4. line 315 and 772, reference 58 should be corrected (DOI: 1158/1078-0432.CCR- 14-1452)

Response: We thank reviewer #2 for the observations; we have corrected them. The change is highlighted in the bibliography using track changes.

Point 5. line 324, space between 15 and kDa.

Response: We thank reviewer #2 for the observations; we have corrected them. The change is highlighted in the main text using track-changes.

Point 6. line 389-395, inconsistency of space with time

Response: Response: We thank reviewer #2 for the observations; we have corrected them. The changes are highlighted in the main text using track-changes. In addition we changed ß+ max by ß+ mean for consistency with figure 4.

Point 7. line 346 and 395, 46Sc should be 44Sc

Response: We thank reviewer #2 for the observationss; we have corrected them. The changes are highlighted in the main text using track-changes.

Point 8. line 401, reference 70 by Cal-Conzalez, J. et al introduces a concept of mPET and several radionuclides, but 44Sc is not included. One of below publications or both introducing 44Sc could be additionally

Rosar, F. et al. Image quality analysis of 44Sc on two preclinical PET scanners: a comparison to 68Ga. EJNMMI Phys. 7, (2020). https://doi.org/10.1186/s40658- 020-0286-3

Rosar, F. et al. Impact of prompt gamma emission of 44Sc on quantification in preclinical and clinical PET systems. Appl. Radiat. Isot. 170, 1–8 (2021). https://doi.org/10.1016/j.apradiso.2021.109599

Response: We apologize for the mistake, and we appreciate the reviewer's indications. We have corrected, and we included the publications indicated by the reviewer (as references 79 and 80). Reference numbers from there have been updated automatically.

The changes are highlighted in the main text using track-changes.

Table 2, 4th row and model column, please check bold

Response: We highlight this refence intentionally as this is the only immuno-PET trazer that reached clinical trials in humans. We have indicated this in the table caption (Antibody-based PET imaging probes reaching clinical trials are highlighted in bold). We can remove bold and this note if the reviewer/editor considers it unnecessary.

* line 503 and 618, 89Zr- à [89Zr]Zr-

Response: We thank reviewer #2 for the observations; we have corrected them. The change is highlighted in the main text using track-changes.

line 589, please check reference 2.

Response: We thank reviewer #2 for the observations; we have corrected them.  Reference has been updated as follows:

Altekruse, S.F.; Kosary, C.L.; Krapcho, M.; Neyman, N.; Aminou, R.; Waldron, W.; Ruhl, J.; Howlader, N.; Tatalovich, Z.; Cho, H.; Mariotto, A.; Eisner, M.P.; Lewis DR, Cronin K, Chen, H.S.; Feuer, E.J.; Stinchcomb, D.G.; Edwards, B.K. (eds). SEER Cancer Statistics Review, 1975-2007, National Cancer Institute. Bethesda, MD, https://seer.cancer.gov/csr/1975_2007/, based on November 2009 SEER data submission, posted to the SEER web site, 2010

* line 728,124I à 124I

Response: We thank reviewer #2 for the observations; we have corrected them. The change is highlighted in the main text using track-changes.

line 804-807, reference

Response: We thank reviewer #2 for the observations; we have corrected them. The change is highlighted in the main text using track-changes.

We appreciate the changes suggested by the reviewer that clearly strengthen this manuscript.

Sincerely,

Alberto J. Schuhmacher
